# The Association between Circulating Cytokines and Body Composition in Frail Patients with Cardiovascular Disease

**DOI:** 10.3390/nu16081227

**Published:** 2024-04-20

**Authors:** Ilona Korzonek-Szlacheta, Bartosz Hudzik, Barbara Zubelewicz-Szkodzińska, Zenon P. Czuba, Patryk Szlacheta, Andrzej Tomasik

**Affiliations:** 1Department of Cardiovascular Disease Prevention, Faculty of Public Health in Bytom, Medical University of Silesia, Piekarska 18, 41-902 Bytom, Poland; ikorzonek@sum.edu.pl (I.K.-S.); bhudzik@sum.edu.pl (B.H.); 2Third Department of Cardiology, Faculty of Medical Sciences in Zabrze, Medical University of Silesia, M. Skłodowskiej-Curie 9, 41-800 Zabrze, Poland; 3Department of Nutrition-Related Disease Prevention, Faculty of Public Health in Bytom, Medical University of Silesia, Piekarska 18, 41-902 Bytom, Poland; bzubelewicz-szkodzinska@sum.edu.pl; 4Department of Microbiology and Immunology, Faculty of Medical Sciences in Zabrze, Medical University of Silesia, Jordana 19, 41-808 Zabrze, Poland; zczuba@sum.edu.pl; 5Department of Basic Medical Sciences, Faculty of Public Health in Bytom, Medical University of Silesia, Piekarska 18, 41-902 Bytom, Poland; 6II Department of Cardiology in Zabrze, Faculty of Medical Sciences in Zabrze, Medical University of Silesia, 40-055 Katowice, Poland; atomasik@sum.edu.pl

**Keywords:** frailty, Fried frailty scale, fat-free mass, cytokines, cardiovascular disease, frailty pathophysiology, risk prediction

## Abstract

The burden of cardiovascular disease and the percentage of frail patients in the aging population will increase. This study aims to assess the circulating levels of several cytokines in frail patients. This is an ancillary analysis of the FRAPICA trial. The ratio of men/women changed from robust through frail groups from 3:1 to 1:2. The groups are comparable in terms of age and body measurements analysis (weight, height, and BMI), yet the frail patients have significantly reduced fat-free mass, and more often have been diagnosed with diabetes. Frail patients have higher fibroblast growth factor basic (FGF basic) and follistatin levels (borderline significance). In multiple linear regression modeling of fat-free mass, we identified FGF basic, osteopontin, stem cell factor, soluble suppression of tumorigenicity 2, soluble epidermal growth factor receptor, soluble human epidermal growth factor receptor 2, follistatin, prolactin, soluble interleukin 6 receptor alfa, platelet endothelial cell adhesion molecule 1, soluble vascular endothelial cell growth factor receptor 1, leptin, soluble angiopoietin/tyrosine kinase 2, and granulocyte colony-stimulating factor. We have identified a few cytokines that correlate with fat-free mass, a hallmark of frailty. They comprise the kinins implicated in bone and muscle metabolism, fibrosis, vascular wall function, inflammation, endocrine function, or regulation of bone marrow integrity.

## 1. Introduction

Cardiovascular disease (CVD) remains the most common cause of morbidity and mortality globally and substantially contributes to loss of health and excess health system costs [1]. Worldwide, the estimated number of deaths due to CVDs increased from around 12.1 million in 1990 to 18.6 million in 2019. In all regions, atherosclerotic CVD (ASCVD)—coronary artery disease and stroke—is the leading cause of CVD mortality [2]. Heart failure (HF) is a rapidly growing public health problem with an estimated prevalence of >60 million patients worldwide. Likewise, it is characterized by significant morbidity, poor prognosis, functional capacity and quality of life, and high healthcare costs [3,4].

The aging process and the coexistence of several comorbid conditions that often conversely interact to yield a greater than additive negative effect on health status is fundamental in the development of frailty syndrome. It is defined as a loss of functionality resulting in increased susceptibility to adverse stress and health events or as a medical syndrome that is described as decreased strength, endurance, and diminished physiologic function that increases a person’s vulnerability for dependency, loss, and/or death [5,6]. The spectrum of frailty varies from robust (not frail), to pre-frail, and physically frail. Frailty is linked to elevated cardiovascular morbidity and mortality both in patients with or without established CVD [6]. The reported frailty prevalence varies greatly depending on the studied population, clinical setting, and the measures used to identify frail persons. A meta-analysis found that the overall prevalence of frailty was 18% (ranging from 12% in 53 community-based studies to 45% in 15 non-community-based studies) [7]. Frailty is highly prevalent among older patients with CVD—women are more affected (approximately 1.6 times) by it than men—especially those with heart failure (HF) (up to 80% of patients) and aortic valve disease (up to 74% of patients). It is linked to a 2.5–3.5-fold elevated mortality risk, even in patients with less severe forms of CVD [8].

Reduced muscle mass—sarcopenia—is considered a key feature of frailty [9]. However, the mechanisms underlying this clinical phenotype are not well defined. There is experimental evidence on the role of carnitine insufficiency in patients with sarcopenia and heart failure or the impact of follistatin on muscle mass in mice is suggested [10]. More often, a multifactorial etiology of frailty or sarcopenia is suggested [11,12]). In general, it is felt to result from a combination of different factors [13]. An improved understanding of the markers and mechanisms of frailty may provide an opportunity for its early detection and for interventions that may prevent its negative outcomes [14].

This study aimed to assess the levels of several circulating cytokines in the plasma of older patients with coronary artery disease and different degrees of frailty and relate the cytokine levels to patients’ lean body mass.

## 2. Materials and Methods

This is an ancillary analysis of the FRAPICA study (ClinicalTrials.org NCT03209414).

We present the analysis of selected plasma cytokines for 78 consecutive patients enrolled in the FRAPICA study. The details of the study were described previously [15].

We use the Fried frailty phenotype score [9] and Lawton and Brody score (Instrumental Activities of Daily Living—IADL). We recognize frailty if three or more out of the five following criteria are met:-Slowness—reduced gait speed at a distance of 5 m at the usual pace. A patient must repeat three times, and the results are averaged. The result is classified as positive if it is >6 s.-Weakness is assessed with a maximal handgrip strength test. It is carried out in the dominant arm. We use an electronic hand dynamometer EH101 (VETEK AB, Sweden). The patient must repeat three times, and the maximal value is recorded. The test is positive for frailty if it is <30 kg for men and <20 kg for women.-Low physical activity is assessed by the Minnesota Leisure Time Activity questionnaire. The result is positive when calorie expenditure per week is lower than 270 kcal/week in women and less than 383 kcal/week in men. We have prepared a Microsoft Excel-based template for rapid questioning and easy calculation of all activities and respective calorie expenditures. We are assessing the physical activity from the recent twelve years.-Exhaustion is self-reported by a patient. The patient has to answer the following questions from the CESD-R scale: “How often in the past week did you feel like everything you did was an effort? How often in the past week did you feel like you could not get going?” The possible answers are often (3 or more days) or not often, when the feeling is present in 0–2 days. The positive answer is when the patient says “often”.-The last criterion is unintentional weight loss exceeding 10 pounds (appr. 4.5 kg) in the past year.

Patients in whom one or two criteria are present will be classified as pre-frail. Patients with a score of 0 are classified as robust.

Moreover, patients are measured for fat-free mass using Harpenden’s skinfold caliper and Baty’s body assessment software v. 17 (Baty International Ltd., Burgess Hill, UK). Lean body mass is derived from a patient’s height and weight using three site Jackson/Pollock algorithm [16].

Patients were enrolled into three groups: (1) robust—20 pts, (2) pre-frail—43 pts, and (3) frail—15 pts. For this analysis, we have used the data of the first 78 consecutively recruited patients, who were admitted to II Department of Cardiology in Zabrze, between May 2017 and December 2018. All patients have signed informed consent for participation in the study. All of the patients have undergone coronary angiography (preferred radial access) and completed a study protocol evaluation. The data on concomitant diseases were retrieved either directly from patients or from patients’ electronic medical records.

### 2.1. Cytokines Analysis

We drew eight milliliters of fasting blood for the preparation of plasma, which was stored at −85 °C for further analysis. Soluble ST2 was measured using the Aspect-Plus ST2 Test (Critical Diagnostics, San Diego, CA, USA).

### 2.2. Evaluation of the Cytokines’ Concentrations

The cytokines’ concentrations were measured using xMAP suspension array technology and all procedures were performed in accordance with the producer’s manuals (Bio-Plex Pro™ Human Cytokine Assay, Bio-Rad Laboratories Inc., Hercules, CA, USA). Briefly, magnetic beads with different shades of red, assigned to individual cytokines, were incubated with samples and standards in wells of microplate. Then, after washing, the biotinylated antibody mixture was added to the wells and incubated. After another washing of the beads, streptavidin–phycoerythrin conjugate was added to the wells and incubated. Then, after the last washing, the beads are suspended in a buffer for further analysis. Washing was performed using an ELx 50 magnetic washer (BioTek, Shoreline, WA, USA) and quantitative measurements of cytokines were analyzed using the Bio-Plex 200 System controlled by Bio-Plex Manager software (Standard Edition, Bio-Rad Laboratories Inc., Hercules, CA, USA) [17,18,19,20,21].

### 2.3. Statistical Analysis

After analysis of the data for normality of distribution and equity of variances, we used the non-parametric statistical methods for comparison of the robust, pre-frail, and frail groups. We have used Kruskal–Walis ANOVA for the comparison of multiple groups, the Mann–Whitney U test for the comparison of two groups, and Chi^2^ with Yates correction for the comparison of frequency data. We have presented data as median and interquartile range or as frequency data. For multiple regression, we have log transformed the data and have used a forward stepwise approach. We have checked with ANCOVA the patients’ gender and presence of diabetes as confounding variables and have additionally performed multiple linear regression modeling for subgroups of males/females, as well as for diabetic/non-diabetic patients [22,23]. For statistical analysis, we used Statistica 13 (Statsoft, PL, Kraków, Poland) licensed to the Medical University of Silesia. The correlogram was prepared in a corrplot package for R version 3.1.0 (the R Foundation for Statistical Computing).

## 3. Results

### 3.1. Characteristics of Frail Population

Patient demographics are presented in Table 1. There is a visible consistent trend in reducing the percentage of men from robust through frail groups from 3:1 to 1:2. Although the groups are comparable in terms of age, co-morbidities, and body measurements analysis (weight, height, and BMI), the frail patients have significantly reduced fat-free mass, and more often have diagnosed diabetes.

### 3.2. The Analysis of Cytokines

The results of cytokines analysis are shown in Table 2. The concentrations of most of the examined cytokines were comparable across the studied groups. There is a slight increasing trend (with borderline significance) in the concentration of basic fibroblast growth factor (FGF basic) and follistatin, with the highest values in frail patients. The cytokines are diffusely cross-correlated and the correlation matrix is presented in the correlogram (Figure 1) and in the Table 3. 

### 3.3. Predictors of Reduced Fat-Free Mass

We consider the reduced fat-free mass as a synonym for sarcopenia. The sarcopenia is a decrease (weakness) in muscle strength, and loss of skeletal muscle mass and their functions. Thus, we have used the correlation matrix of cytokines to model the fat-free mass with multiple linear regression analysis. As patient’s gender and diabetes were identified as confounding variables, we have performed separate sub-analyses for men/women and diabetic/non-diabetic patients. The results are presented in Table 4. Fat-free mass in the entire study population is best described by FGF basic, osteopontin, SCF, sST2, sEGFR, and sHER-2new with multiple R 0.48. The separate analysis in men has confirmed the FGF basic and osteopontin as predictors of lean body mass; however, the sub-analysis in women has disclosed a completely different panel of predicting cytokines. The analysis in diabetic/non-diabetic patients has confirmed the negative correlation between FGF basic and lean body mass.

## 4. Discussion

Our study population represents the typical characteristics of progressing frailty, from the robust state, through pre-frailty to regular frailty with an increasing proportion of women and significantly reduced fat-free mass. Moreover, frail patients more frequently have diabetes. The analysis of cytokines has disclosed different correlation matrices of fat-free mass depending on the patient’s gender and diabetic status. Increased concentration of fibroblast growth factor (FGF) basic was the independent variable associated with reduced lean body mass in the whole study population and male and diabetic subgroups. It was also a component of an equation describing the fat-free mass in the non-diabetic subgroup. Several publications have addressed the association of inflammatory, coagulation, and hormonal factors with frailty development [24,25,26,27].

Beben et al. [28], in a cross-sectional observational study, have found that higher fibroblast growth factor FGF23 plasma concentrations are independently associated with frailty. Our study patients have a similar trend in FGF basic concentrations with the highest plasma levels in frail patients (borderline significance). Yet, the FGF basic was the primary variable to describe the negative correlation with fat-free mass in our patients. Beben et al. have analyzed the odds of individual frailty traits by doubling FGF23 concentration and the odds ratio for weight loss was 1.22. Another study by Holecki et al. [29] has reported on the lack of association between obesity and plasma concentration of FGF23 in patients aged 65 or more without mentioning patients’ frailty. Both studies, either that by Beben et al. [28] or ours, have the cross-sectional approach, which has the least predictive power to infer the risk and causality. Yet, Beben et al. have suggested that FGF-23 may serve as an early marker of frailty risk that may be used to institute therapeutic interventions.

Follistatin is a myostatin-binding protein that inhibits myostatin activity and promotes muscle growth [10]. We have not measured in our experiment the concentration of myostatin, so we cannot explain why our frail patients, despite the higher concentration of follistatin, have lower muscle mass. Echevarria et al. [30], in a similarly small study, have found that frail patients (Fried phenotype) have significantly higher follistatin concentrations. However, they failed to correlate the increased follistatin concentration with sarcopenia defined according to the revised European consensus [31]. Echevarria et al., in their binary model, have found insignificantly higher follistatin levels in sarcopenic patients in comparison with patients with normal appendicular skeletal muscle mass and we have used the linear approach to model the fat-free mass [30].

Leptin has a dual impact on fat-free mass in our population. As it is not a predictor of FFM in our general population, it is correlated positively with FFM in women, and negatively in diabetic patients. Hubbard et al. [32], in their cross-sectional study, found that the most frail patients have significantly lower leptin levels and the leptin concentration correlated positively with fat mass, expressed as triceps skinfold thickness. A study by Kohara et al. [33] on sarcopenic obesity may serve as an explanation for the observation by Hubbard et al. [32]. Other evidence on the role of leptin in frailty development is provided by Lana et al. [34], who have found that higher serum leptin levels lead to the development of incident frailty in a two-year-long follow-up.

The osteopontin concentrations correlated positively with the fat-free mass in our patients, although Boreskie et al. [35] in a large-scale case–control study ruled out the possible impact of plasma osteopontin level on the development of pre-frailty status.

There is no direct evidence in the literature on the impact of the other cytokines that we have found to describe the reduced fat-free mass in an elderly population (SCF, sST2, sEGFR, sHER2-new, prolactin, sIL6-RA, PECAM-1, sVEGRF-1, sTIE-2, and G-CSF) on frailty. However, there is some evidence that further studies are warranted or that some of the above-mentioned cytokines may serve as risk factors. Soluble ST2 in a comorbid frail elderly population with heart failure with preserved ejection fraction outperformed NT-proBNP for predicting the risk of all-cause mortality or heart failure-related rehospitalization [36]. The sEGFR, PECAM-1, sVEGFR-1, and sTIE-2 are implicated in the regulation of fibrotic processes, vascular integrity, and vessel wall homeostasis, considering that most of our patients have been diagnosed with coronary artery disease [37,38]. The sIL-6RA reflects the age-related process of inflammation [25,29], while sHER-2new, as well as prolactin, may reflect the endocrine dysfunction in an elderly population.

Eventually, studies on colony-stimulating factor CSF, or more accurately, stem cells and granulocyte-colony stimulating factor G-CSF, are extremely promising in skeletal muscle regeneration [39] or even treatment of frailty [40,41].

We have not studied the dietary patterns and habits of our patients, nor have we assessed blood analytes for post-prandial changes. However, there is evidence in the literature that a single high-fat meal increases tumor necrosis factor α (TNF-α) concentration in patients with metabolic syndrome [42]. This is an intriguing observation, especially in the context of a population with established cardiovascular disease or sarcopenia. Another issue we have to underline here is the fact that cardiovascular disease by itself might modify the cytokine levels.

### 4.1. Study Limitations

First, we failed to indicate directly the associations of studied cytokines and frailty, yet we have found a few of them that describe the reduced fat-free mass as the hallmark of frailty. This reflects the complex pathophysiology of incident frailty. Second, the cross-sectional design of our study has the least predictive power to infer the risk and causality [18]. Third, our study group is relatively small and this limits the statistical capability further. The sample size analysis, based on FGF basic estimates, has yielded 45 patients in each of the three groups and the comparable number of patients is only in the pre-frail subgroup. In the remaining groups, we have half and a third of the required number of patients. Moreover, we have assumed the linear distribution of the variables across the subgroups, yet a small sample size does not allow us to exclude non-linear distribution and any residual effect of outlier results. It is important to stress that some authors advocate that changes at the physiological and molecular levels in frail patients might have non-linear dynamics [13]. Nonetheless, some of our results are consistent with the previously published paper, while other cytokines require a more precisely designed investigation to elucidate their pathophysiological impact on frailty and eventually evaluate their prognostic utility.

### 4.2. Future Directions

The concept of frailty, which subsumes a diversity of vulnerabilities, weaknesses, instabilities, or limitations, was raised during the 2004 American Geriatrics Society/National Institute on Aging (AGS/NIA) Conference on a Research Agenda on Frailty in Older Adults, as described by Walston et al. [13], and was also recently documented in practice by Marcucci et al. [43]. The combination of clustering of clinical data with many biomolecular variables seems reasonable to provide us with a better understanding of factors predisposing to the progression of frailty and will allow us to tailor the care and management depending on patients’ needs.

## 5. Conclusions

We have identified a few cytokines that correlate with fat-free mass—a hallmark of frailty. They comprise the kinins implicated in bone and muscle metabolism, fibrosis, vascular wall function, inflammation, endocrine function, or regulation of bone marrow integrity. This reflects the complex nature of the frailty syndrome.

## Figures and Tables

**Figure 1 nutrients-16-01227-f001:**
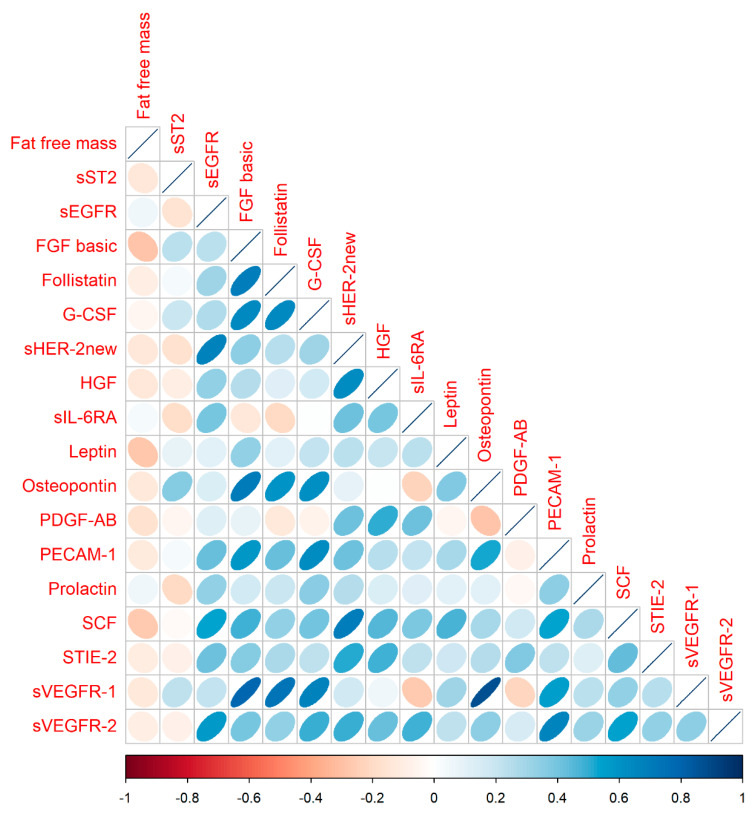
Correlogram of studied cytokines. The blue ovals indicate positive correlation, while the beige ovals indicate the negative correlations. The thinner the oval is the stronger the correlation.

**Table 1 nutrients-16-01227-t001:** Patient demographics and clinical presentation.

	All Patients (*n* = 78)	Robust (*n* = 20)	Pre-frail (*n* = 43)	Frail (*n* = 15)	Significance (Kruskal–Wallis ANOVA)
Age, years, median (IQR)	69.0 (9.0)	69.0 (5.5)	68.0 (11.0)	72.0 (9.0)	*p* = 0.458
Gender, M/F, *n*	49/39	15/5	29/14	5/10	*p* < 0.05
Height, m, median (IQR)	1.700 (0.14)	1.725 (0.14)	1.700 (0.14)	1.640 (0.13)	*p* = 0.125
Weight, kg, median (IQR)	81.0 (19.0)	84.50 (22.50)	78.000 (17.00)	82.00 (32.00)	*p* = 0.324
BMI, kg/m^2^, median (IQR)	28.4 (5.4)	29.1 (5.7)	27.8 (4.6)	30.7 (10.0)	*p* = 0.253
Fat-free mass, kg, median (IQR)	53.00 (14.00)	58.65 (16.05)	53.00 (12.4)	42.1 (9.7)	*p* < 0.001
Coronary artery disease, *n* (%)	73 (94)	19 (95)	41 (95)	13 (87)	*p* = 0.48
Hypertension, *n* (%)	66 (85)	17 (85)	37 (86)	12 (80)	*p* = 0.856
Diabetes, *n* (%)	33 (42)	7 (35)	14 (33)	12 (80)	*p* < 0.05
Chronic obstructive pulmonary disease, *n* (%)	12 (15)	3 (15)	7 (16)	2 (13)	*p* = 0.466
Chronic renal failure, *n* (%)	4 (5)	0 (0)	2 (5)	2 (13)	*p* = 0.303
Smoking, *n* (%)	21 (27)	6 (30)	13 (30)	2 (13)	*p* = 0.712

**Table 2 nutrients-16-01227-t002:** The analysis of cytokines in subgroups of patients. Data presented are median and interquartile range.

	All Patients (*n* = 78)	Robust (*n* = 20)	Pre-Frail (*n* = 43)	Frail (*n* = 15)	Significance (Kruskal–Wallis ANOVA)
sST2, ng/mL	25.688 (18.42)	30.566 (15.59)	23.879 (16.01)	33.450 (26.34)	*p* = 0.065
sEGFR	8011.630 (7258.03)	8668.745 (8376.51)	7760.650 (7410.95)	7907.200 (4554.41)	*p* = 0.959
FGF basic	89.150 (52.17)	78.835 (53.42)	88.320 (56.07)	104.130 (77.44)	*p* = 0.0546
Follistatin	273.730 (306.67)	177.865 (197.90)	280.580 (323.29)	464.420 (372.58)	*p* = 0.0714
G-CSF	81.320 (33.92)	76.450 (36.22)	81.110 (33.12)	94.780 (59.54)	*p* = 0.46
sHER-2new	3163.665 (2406.52)	3095.760 (2310.75)	3058.740 (2629.98)	3464.110 (2505.51)	*p* = 0.655
HGF	693.280 (485.28)	665.595 (538.96)	696.520 (574.41)	736.210 (777.58)	*p* = 0.627
sIL-6RA	7667.835 (3166.27)	6755.965 (2182.74)	8038.640 (3672.75)	7965.730 (2834.65)	*p* = 0.352
Leptin	4626.890 (10,153.93)	4405.740 (6743.39)	4418.710 (10,897.44)	6708.200 (10,519.41)	*p* = 0.486
Osteopontin	4231.720 (7736.82)	4209.705 (11,171.82)	2884.290 (7736.82)	6533.970 (41,661.13)	*p* = 0.204
PDGF-AB	898.420 (1709.68)	742.400 (1683.09)	794.120 (1594.41)	1408.680 (1872.71)	*p* = 0.925
PECAM-1	4453.135 (1815.60)	4151.790 (2047.52)	4556.280 (1821.25)	4797.900 (2026.70)	*p* = 0.161
Prolactin	3941.705 (4131.73)	2931.650 (4686.14)	4075.620 (3759.22)	3697.980 (4342.15)	*p* = 0.877
SCF	107.500 (60.20)	85.215 (79.39)	110.870 (54.70)	120.470 (61.12)	*p* = 0.194
sTIE-2	2836.785 (3001.54)	3153.260 (2941.08)	2642.790 (3197.91)	4077.990 (2826.10)	*p* = 0.306
sVEGFR-1	66.865 (170.09)	50.355 (128.63)	70.170 (139.73)	90.050 (726.83)	*p* = 0.249
sVEGFR-2	641.525 (489.23)	619.960 (416.69)	636.410 (512.71)	751.930 (655.89)	*p* = 0.394

**Table 3 nutrients-16-01227-t003:** The correlation matrix of fat-free mass and studied cytokines. Presented are R values; data in red are significant at *p* < 0.05; minus is for negative correlation.

	Fat Free Mass	sST2	sEGFR	FGF Basic	Follistatin	G-CSF	sHER-2new	HGF	sIL-6RA	Leptin	Osteopontin	PDGF-AB	PECAM-1	Prolactin	SCF	S-TIE2	sVEGFR-1	sVEGFR-2
Fat-free mass																		
sST2	−0.13																	
sEGFR	0.05	−0.16																
FGF basic	−0.28	0.23	0.24															
Follistatin	−0.09	0.03	0.31	0.69														
G-CSF	−0.04	0.20	0.27	0.65	0.64													
sHER-2new	−0.13	−0.16	0.67	0.35	0.24	0.30												
HGF	−0.14	−0.09	0.33	0.26	0.11	0.16	0.64											
sIL-6RA	0.04	−0.18	0.39	−0.13	−0.21	0.00	0.42	0.39										
Leptin	−0.27	0.09	0.10	0.33	0.10	0.21	0.23	0.20	0.23									
Osteopontin	−0.12	0.37	0.13	0.70	0.60	0.61	0.10	0.00	−0.22	0.38								
PDGF-AB	−0.16	−0.05	0.13	0.09	−0.12	−0.07	0.42	0.51	0.42	−0.04	−0.28							
PECAM-1	−0.11	0.03	0.42	0.58	0.42	0.64	0.42	0.25	0.21	0.28	0.53	−0.07						
Prolactin	0.05	−0.20	0.34	0.17	0.19	0.35	0.26	0.13	0.12	0.11	0.10	−0.04	0.35					
SCF	−0.26	−0.02	0.54	0.48	0.33	0.41	0.69	0.45	0.38	0.48	0.29	0.17	0.54	0.27				
S—TIE2	−0.10	−0.07	0.42	0.37	0.28	0.23	0.52	0.48	0.23	0.18	0.25	0.37	0.23	0.13	0.43			
sVEGFR-1	−0.12	0.23	0.22	0.80	0.73	0.67	0.17	0.06	−0.27	0.31	0.88	−0.22	0.55	0.24	0.34	0.25		
sVEGFR-2	−0.09	−0.08	0.57	0.40	0.33	0.50	0.49	0.42	0.48	0.23	0.35	0.15	0.67	0.31	0.55	0.34	0.34	

**Table 4 nutrients-16-01227-t004:** Multiple linear regression analysis of predictors of fat-free mass.

Group/Subgroup of Patients	Adjusted Variable	Predicting Variables	Β	t	*p*
All patients		FGF basic	−0.38	−2.39	0.019
		Osteopontin	0.27	1.87	0.065
		SCF	−0.19	−1.11	0.271
		sST2	−0.16	−1.44	0.153
		sEGFR	0.22	1.57	0.120
		sHER-2new	−0.21	−1.20	0.235
Regression summary		Multiple R = 0.48 Multiple R^2^ = 0.24 Corrected R^2^ = 0.17 F(6.71) = 3.59 *p* < 0.01			
Adjusted for sex	Male	FGF basic	−0.49	−2.77	0.008
		Osteopontin	0.40	2.29	0.027
Regression summary		Multiple R = 0.39 Multiple R^2^ = 0.15 Corrected R^2^ = 0.11 F(2.46) = 4.07 *p* < 0.05			
Adjusted for sex	Female	Follistatin	−0.58	−2.44	0.024
		Prolactin	0.30	1.831	0.081
		sIL-6RA	−0.69	−3.26	0.004
		PECAM-1	0.38	1.74	0.096
		sVEGFR-1	−0.31	−1.12	0.251
		Leptin	0.31	1.47	0.155
		sTIE-2	−0.22	−1.32	0.203
Regression summary		Multiple R = 0.73 Multiple R^2^ = 0.53 Corrected R^2^ = 0.37 F(7.21) = 3.39 *p* < 0.05			
Adjusted for diabetes	No diabetes	SCF	−0.26	−1.13	0.265
		sST2	−0.35	−2.34	0.024
		Osteopontin	0.55	2.92	0.006
		FGF basic	−0.42	−2.12	0.04
		sHER-2new	−0.26	−1.14	0.262
Regression summary		Multiple R = 0.58 Multiple R^2^ = 0.33 Corrected R^2^ = 0.25 F(5.39) = 3.88 *p* < 0.01			
Adjusted for diabetes	Diabetes	FGF basic	−0.45	−2.02	0.053
		sEGFR	0.28	1.72	0.097
		Leptin	−0.28	−1.53	0.137
		G-CSF	0.23	1.06	0.296
Regression summary		Multiple R = 0.54 Multiple R^2^ = 0.3 Corrected R^2^ = 0.2 F(4.28) = 2.94 *p* < 0.05			

## Data Availability

The original contributions presented in the study are included in the article, further inquiries can be directed to the corresponding authors.

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
