# Peer review of "The Association between Circulating Cytokines and Body Composition in Frail Patients with Cardiovascular Disease"

_nutrients, 2024, doi:10.3390/nu16081227_

Round 1

Reviewer 1 Report

Comments and Suggestions for Authors

Dear Authors, 

This study aimed to evaluate the levels of different cytokines circulating in the plasma of 78 elderly patients with coronary heart disease, enrolled in the FRAPICA study, according to different degrees of frailty and to correlate these cytokine levels to the patients' lean mass which represents a marker of frailty.

Main criticism

The article is titled "Association between circulating kinins and body composition in frail patients with cardiovascular disease", but the article aims to assess the circulating levels of several cytokines in frail patients. Kinins are vasoactive peptides released from precursors called kininogens, a family of amino acid peptide hormones that have a variety of important physiological and pathological effects. Kinins may induce the production of proinflammatory cytokines. Therefore the authors should change the title by replacing the term kinins with cytokines

The authors should better describe the Frapica study population recruited at moment of this analysis and on what the selection criteria were for this sample of 78 patients. The reported reference of Woloszyn-Horak E et al. Medicine (Baltimore) 2020, described the design of this study, but there is no information on the population sample recruited.

In figure 1 the correlation matrices with statisitical signifcance (p value) should also be shown in addition to the correlogram of the cytokines studied.

The sample size calculation for the cross-sectional study should be performed before doing this analysis because the results obtained are perhaps due to the small size of the sample studied as for example  for the borderline association between degree of frailty and fgf.

This is a cross-sectional study so it is not correct to talk about predictors but the authors should specify that it is an association between cytokines and fat-free mass

Comments on the Quality of English Language

Moderate editing of English language required

Author Response

Dear Authors, 

This study aimed to evaluate the levels of different cytokines circulating in the plasma of 78 elderly patients with coronary heart disease, enrolled in the FRAPICA study, according to different degrees of frailty and to correlate these cytokine levels to the patients' lean mass which represents a marker of frailty.

Main criticism

The article is titled "Association between circulating kinins and body composition in frail patients with cardiovascular disease", but the article aims to assess the circulating levels of several cytokines in frail patients. Kinins are vasoactive peptides released from precursors called kininogens, a family of amino acid peptide hormones that have a variety of important physiological and pathological effects. Kinins may induce the production of proinflammatory cytokines. Therefore the authors should change the title by replacing the term kinins with cytokines

Answer: Thanks a lot for the comment. We have replaced “kinins” with “cytokines”

The authors should better describe the Frapica study population recruited at moment of this analysis and on what the selection criteria were for this sample of 78 patients. The reported reference of Woloszyn-Horak E et al. Medicine (Baltimore) 2020, described the design of this study, but there is no information on the population sample recruited.

Answer: This analysis population comprises first 78 patients recruited into FRAPICA trial between May 2017 and December 2018. The patients were admitted to II Department of Cardiology and all of them underwent coronary angiography and complete trial protocol evaluation. We have made appropriate correction in Method Section.

PS: So far (including very long COVID-19 pandemics break) we have enrolled 494 patients, and do hope to complete recruitment of 1000 patients by end of December 2024.

In figure 1 the correlation matrices with statisitical signifcance (p value) should also be shown in addition to the correlogram of the cytokines studied.

Answer: We have added one more table with detailed correlation matrix of fat free mass and studied cytokines.

The sample size calculation for the cross-sectional study should be performed before doing this analysis because the results obtained are perhaps due to the small size of the sample studied as for example  for the borderline association between degree of frailty and fgf.

Answer: The sample size analysis for FGF basic with borderline significance among three tested groups is 45 in each group, and we have 20 patients in robust group, 43 in pre-frail group, and 15 in frail group. The proportion of robust/pre-frail/frail patients is consistent across the studied papers with abundance of pre-frail patients. When we decided to perform the cytokine analysis we have had limited number of diagnostic kits as well as  limited patients blood samples, to perform more advanced analyses like pairing the patients with similar age, sex, or diversify them with more rigoristic frailty traits selection.  We are aware of small number of cases, and sort of a pilot study nature of our analysis and we have extended The limitation section on these, along with the linearity of data issues raised by Reviewer 2.

This is a cross-sectional study so it is not correct to talk about predictors but the authors should specify that it is an association between cytokines and fat-free mass

Answer: We have made pertinent corrections.

Moderate editing of English language required

Answer: We think MDPI professional language service will provide best result.

Thanks a lot for your comments

Reviewer 2 Report

Comments and Suggestions for Authors

The authors conducted an observational study to examine the associations of cytokines with frailty and lean body mass in patients with cardiovascular diseases. By analyzing the data of 78 patients enrolled in a trial, the authors showed that levels of several cytokines, notably FGF basic, were associated with lean body mass in this study sample. These findings are interesting. There are some comments.

1.      Title: The authors used the term “kinins” in the title. However, its description was lacking in the Introduction and Methods. I recommended either using another term in the Title or defining and describing the term in the rest of the manuscript.

2.      Introduction: “The study aimed to assess the levels of several circulating cytokines in the plasma --.” However, it is unclear why the authors hypothesized an association of these cytokines with frailty or muscle mass. A more detailed description of the rationale is recommended.

3.      Materials and Methods (Statistical analysis): Linear regression was applied. Please describe whether (and how) the assumptions, for instance, constant variance and the absence of outliers’ effects, were checked and were met.

4.      Materials and Methods: The authors describe the measurements of the main exposure (cytokines) and outcome variables (frailty, lean body mass). Please also describe the measurements of variables (for instance, diabetes) that were analyzed as confounders in this study.

5.      Results (Table 3): The authors conducted multiple linear regression analysis and presented the estimated adjusted associations of cytokines with lean body mass. Please also present the unadjusted association estimates.

6.      Discussion: Levels of multiple cytokines tested in this study sample did not vary much across patients with different frailty status. One issue that worth discussion is that this study sample consisted of patients with cardiovascular diseases. Levels of certain cytokines may be universally elevated or decreased in this sample. As such, variations of certain cytokines levels across patients with different frailty status were not observed. A study recruiting larger number of cardiovascular diseases patient may be needed to test (or refute) the associations of these cytokines’ levels with frailty.

7.      Discussion: Although the regression analysis has been adjusted for multiple confounders, residual confounding is still possible in this observational study. A discussion of this issue is suggested.

Author Response

The authors conducted an observational study to examine the associations of cytokines with frailty and lean body mass in patients with cardiovascular diseases. By analyzing the data of 78 patients enrolled in a trial, the authors showed that levels of several cytokines, notably FGF basic, were associated with lean body mass in this study sample. These findings are interesting. There are some comments.

  1. Title: The authors used the term “kinins” in the title. However, its description was lacking in the Introduction and Methods. I recommended either using another term in the Title or defining and describing the term in the rest of the manuscript.

Answer: We have replaced “kinins” with “cytokines”

  1. Introduction: “The study aimed to assess the levels of several circulating cytokines in the plasma --.” However, it is unclear why the authors hypothesized an association of these cytokines with frailty or muscle mass. A more detailed description of the rationale is recommended.

Answer: We have added some more information along with new references on the role physical activity, diet, carnitine, follistatin, and “precision medicine approach”.

  1. Materials and Methods (Statistical analysis): Linear regression was applied. Please describe whether (and how) the assumptions, for instance, constant variance and the absence of outliers’ effects, were checked and were met.

Answer: Thanks a lot for this comment. Both, You and Reviewer 1 have addressed the issue of the power of our analysis. We are completely aware of small sample size (the sample size, based on FGF concentrations, should comprise 45 cases in each of three study groups, and we had 20-43-15 in robust/pre-frail/frail groups respectively. This is why we decided to present the data and analyze them primarily non-parametrically. The linearity of data is only our assumption and we are not able to prove this definitely in our study population. Especially, when we consider the idea of chaos:

“…Discussions of nonlinear dynamics and chaos theory provided new insights into how investigators might detect and quantify changes at the physiological and molecular level. It was suggested that, in vital and resilient organisms, complex physiological pathways allow a wide variety of adaptive responses that are qualitatively and quantitatively modified to specific events. This complexity helps keep multiple systems in balance with minimal fluctuations in the homeostatic equilibrium. Aging results in the decline of normal interactions and the redundancy of communication between these physiological systems. It was hypothesized that frailty, both the phenotype and its latent vulnerability, results from reaching a threshold of decline in one or more systems that triggers a cascade of dysregulation in multiple systems and that this dysregulation may influence many clinical domains, as well as comorbid conditions and disability….”  

Described by Walston et al. (J. Walston, E. C. Hadley, L. Ferrucci, J. M. Guralnik, A. B. Newman, S. A. Studenski, et al.; J Am Geriatr Soc 2006 Vol. 54 Issue 6 Pages 991-1001;

Accession Number: 16776798 DOI: 10.1111/j.1532-5415.2006.00745.x).

To address this issue we have added some comments in Study Limitations section.

  1. Materials and Methods: The authors describe the measurements of the main exposure (cytokines) and outcome variables (frailty, lean body mass). Please also describe the measurements of variables (for instance, diabetes) that were analyzed as confounders in this study.

Answer: We have retrieved the information from patients history, we have recorded the treatment. We have made corrections in manuscript.

PS: Recently, accordingly to ESC Guidelines, patients with concomitant heart failure were advised to receive SGLT2i tablets, and patients with obesity are encouraged to use GLP1 analogs

  1. Results (Table 3): The authors conducted multiple linear regression analysis and presented the estimated adjusted associations of cytokines with lean body mass. Please also present the unadjusted association estimates.

Answer: As we have added new table 3 with cytokine correlation matrix the table You have mentioned is 4 indeed. The unadjusted association is described “All patients”

  1. Discussion: Levels of multiple cytokines tested in this study sample did not vary much across patients with different frailty status. One issue that worth discussion is that this study sample consisted of patients with cardiovascular diseases. Levels of certain cytokines may be universally elevated or decreased in this sample. As such, variations of certain cytokines levels across patients with different frailty status were not observed. A study recruiting larger number of cardiovascular diseases patient may be needed to test (or refute) the associations of these cytokines’ levels with frailty.

Answer: Thanks a lot for this comment. The FRAPICA trial primary objective was to assess the frail cardiovascular patients outcomes in three-year-long follow-up, based on telephone contact, and we have not planned the comparator group, eg. Patients without coronary atherosclerotic stenoses. But, as we are sampling patients blood and keep it frozen, and we continue patient recruitment (we have 494 patients so far, and plan to complete 1000 pts by the end of December 2024) we will enroll a group of sex and age matched coronary artery disease free patients. We have discussed the issue of CVD impact on the cytokine levels in manuscript

  1. Discussion: Although the regression analysis has been adjusted for multiple confounders, residual confounding is still possible in this observational study. A discussion of this issue is suggested.

Answer: We have made manuscript corrections.

Thanks a lot for Your comments.

Reviewer 3 Report

Comments and Suggestions for Authors

The manuscript is interesting and well written. The conclusions are supported by the results. The figures and tables are clear. The limitations of the study are exhaustively described by the authors.

This reviewer raises only a few issues.

1- The role of cytokines, in particular pro-inflammatory, should be studied not only to fasting but also after a meal. In fact, the high-fat meal produces increases in the TNF-alpha levels associated with endothelial dysfunction with a consequent increase in the Cv risk (Nutrition, Metabolism and Cardiovascular Diseases. 2007, 17 (4) risk: 274 - 279. Doi : 10, 10, 10, 10 1016 /J.Numecd. 2005.11. 014). This intriguing and clinically relevant issue should be addressed by the authors under discussion, also considering that TNF-alpha has not been assessed in this study.

2- Using alternative analytical techniques among the oldest hospitalized patients, it is possible to distinguish several phenotypes of fragility, in a different way associated with adverse events. The identification of different patient profiles can help to define the best assistance strategy according to the specific needs of patients (Journals of Gerontology - Series A Biological Sciences and Medical Sciences. 2017. 72(3): 395 - 402. doi: 10.1093/gerona/glw188). This issue needs to be commented in discussion.

Comments on the Quality of English Language

Minor editing of English language is required.

Author Response

The manuscript is interesting and well written. The conclusions are supported by the results. The figures and tables are clear. The limitations of the study are exhaustively described by the authors.

This reviewer raises only a few issues.

1- The role of cytokines, in particular pro-inflammatory, should be studied not only to fasting but also after a meal. In fact, the high-fat meal produces increases in the TNF-alpha levels associated with endothelial dysfunction with a consequent increase in the Cv risk (Nutrition, Metabolism and Cardiovascular Diseases. 2007, 17 (4) risk: 274 - 279. Doi : 10, 10, 10, 10 1016 /J.Numecd. 2005.11. 014). This intriguing and clinically relevant issue should be addressed by the authors under discussion, also considering that TNF-alpha has not been assessed in this study.

Answer: Thanks a lot for this suggestion. We have read the manuscript with interest and made corrections in Discussion. 

2- Using alternative analytical techniques among the oldest hospitalized patients, it is possible to distinguish several phenotypes of fragility, in a different way associated with adverse events. The identification of different patient profiles can help to define the best assistance strategy according to the specific needs of patients (Journals of Gerontology - Series A Biological Sciences and Medical Sciences. 2017. 72(3): 395 - 402. doi: 10.1093/gerona/glw188). This issue needs to be commented in discussion.

Answer: Thanks a lot for this suggestion. We have added a section “Future directions” to indicate the possibility of utilization of clustering clinical data with introduction of biomolecular analyses to define the risk, the care strategy depending on the patients needs.  

Minor editing of English language is required.

Answer: We will ask MDPI Editorial Language service for professional assistance.

Round 2

Reviewer 1 Report

Comments and Suggestions for Authors

Dear Authors,

Thank you for taking my comments into account in this revision of your article. As already reported in the previous review, the work is interesting, but unfortunately the small number of the analysed sample only allows preliminary conclusions to be drawn.

Comments on the Quality of English Language

Minor editing of English language required